# Identification and Characterization of Injuries during Competition in Wheelchair Basketball

**Karina Sá** [1] , **Marília Magno e Silva** [2] , **José Gorla** [1] and **Anselmo Costa e Silva** [2,*]

1 Adapted Physical Assessment, Exercise and Sport Laboratory, Physical Education College, State University of Campinas, Campinas 13083-851, Brazil; jigorla@uol.com.br (J.G.)
2 Adapted Physical Activity Laboratory, Graduate Program in Human Movement Sciences, Federal University of Pará, Belém 66075-110, Brazil
* Correspondence: anselmocs@ufpa.br

**Abstract:** Background: Knowledge and understanding of the most diverse aspects surrounding the emergence of sports injuries stand out as one of the pillars for sporting success. Methods: A total of 41 athletes answered an online form based on the Sports Injury Protocol in Paralympic Sports (PLEEP) in which data on sports injuries during competition in the first Brazilian wheelchair basketball division were collected. Results: The athletes who composed the sample perform a high volume of training. The majority did not present with injuries before the competition. There was a prevalence of 17.1% of injuries during the competition, an incidence of 0.17 injuries per athlete and an incidence rate of 0.03 injuries per athlete-hour or four injuries per one thousand athletes-days. The injuries that occurred during the competition were mostly in the shoulder region, characterized as traumatic, by indirect contact with other athletes, which occurred when propelling the wheelchair, and were of low severity. Conclusions: The wheelchair basketball athletes presented a low prevalence, incidence and incidence rate of injuries during the competition. The shoulder region was the most affected. Structuring training sessions with the prevention of injuries in the shoulder region in mind is essential for these athletes to perform optimally.

**Keywords:** prevalence; incidence; athletic injuries; para-athletes

## 1. Introduction

The World Health Organization (WHO) defines health as a state of complete physical, mental and social well-being, not just the absence of disease or infirmity [1]. Based on this assumption, health problems caused directly or indirectly by sports practice can have different outcomes. Over the years, many authors have tried to define what sports injuries are; therefore, in the year 2020, the consensus of the International Olympic Committee declared injuries to be "tissue damage or other disturbance of normal physical function due to participation in sports, resulting from the rapid or repetitive transfer of kinetic energy" and defined illness as "a complaint or disturbance experienced by an athlete, not related to injury. Illnesses include problems related to physical (e.g., influenza), mental (e.g., depression) or social well-being, or removal or loss of vital elements (air, water, heat)" [2].

These definitions refer to unfavorable conditions that can be experienced by athletes when practicing some sports. Injuries bring with them negative physical consequences [3] such as pain, which often prevent the athlete from practicing sports, and also negative consequences of a psychological nature [4,5], such as anxiety, depression and fear of reinjury. These factors can negatively impact the recovery and consequent return of the athlete to the modality. Fear of reinjury, for example, relates to the increased recovery time of physical impairments, reduces the self-report function and prevents a successful return to sports [6], and for these cases, not only physical approaches are important but also psychological approaches.

In this sense, surveillance of injuries and illnesses is a fundamental element to protect the health of athletes and optimize their performance [2]. This health surveillance goes through the collection, storage and analysis by the clubs/teams and technical commission of the appearance of injuries and related factors, and thus, they are able to outline strategies aimed at optimizing training and preserving the athlete's health so that the athlete can reach their optimal point of performance. In that sense, most sports injuries can be controlled and avoided by implementing preventive measures [7].

When it comes to sports injuries, when practicing any modality, the athlete is exposed to the appearance of injuries, and this is no different in sports for people with disabilities. In addition to exposure to the modality, it is necessary to take into account pre-existing conditions arising from the most different disabilities [8]. In a study that aimed to describe the prevalence of the one-year retrospective period and the point prevalence of sports injuries in Swedish athletes with disabilities, it was identified that the prevalence of severe injuries in the one-year period was 31% (95% CI: 23–40), and the point prevalence was 32% (95% CI: 24–41). The most serious injuries ($p < 0.05$) were reported by athletes between 18 and 25 years of age who were not using an auxiliary device with pain during the sport, using analgesics, continuing injured training and feeling guilty for missing training exercise. In addition, they demonstrated that being female, reporting a previous serious illness, using prescribed medication and feeling anxious/depressed were characteristics associated with ongoing illness ($p < 0.05$) [9].

Therefore, knowledge and understanding of the most diverse aspects surrounding the emergence of sports injuries stand out as one of the pillars for sporting success. In addition, given the specific characteristics related to the disabled athlete, this study aims to identify and characterize sports injuries that occurred during a competition in the first Brazilian wheelchair basketball division.

## 2. Materials and Methods

### 2.1. Characterization of the Study

This study is characterized as an epidemiological cohort. The research protocol was approved by the Research Ethics Committee, and all participants signed the informed consent form. The structure of this research was based on the items described in Strengthening the Reporting of Observational Studies in Epidemiology, Extension for Sports Injury and Illness Surveillance (STROBE—SIIS) [2]. Data collection for this study took place during the Brazilian Men's Championship of the First Division of Wheelchair Basketball, which took place in São Paulo from 11 October 2022 to 17 October 2022.

### 2.2. Sample

During the competition, the athletes, coaches and technical managers of the teams were approached and introduced to the research. For those who agreed to participate in the study, an online form was sent. Data were collected through online forms based on the Sports Injury Protocol in Paralympic Sport (PLEEP) [10] and reported by the athletes and coach during the 7 days of competition. The eligibility criteria to participate in the survey were male players aged over 18 years. The main variables collected were gender, type of disability, disability diagnosis, sports class, occurrence of injury in competition, exposure period, injury mechanism, injury diagnosis, lesion location and new or recurrent injury.

### 2.3. Definition and Characterization of Sports Injuries

For this work, a sports injury was defined as "any tissue damage or other disturbance of normal physical function due to participation in sports, resulting from the rapid or repetitive transfer of kinetic energy" [2]. The mode of onset of injury can be acute (sudden onset) or repetitive (sudden or gradual onset). The injury mechanism can be caused by no contact or by contact; this contact can be direct (with another athlete or object) or indirect (through another athlete or object).

The measure of incidence was defined as the number of new lesions in the population that develop during a defined period [2]. The incidence of the result is presented by injuries/athlete/time of competition and injuries/1000 athletes/day. The prevalence was obtained by dividing the number of existing cases by the total population at risk at a given time [2].

*2.4. Data Analysis*

Data were tabulated in Excel, and all equations for calculating the incidence and prevalence were performed in this program. Data were presented with descriptive statistics showing sum, mean, standard deviation and percentages. The frequency was calculated for variables such as gender, sports class, cause of disability and characteristics. For the calculation of the athletes' exposure, we considered that the teams played 7 games in total. Each start takes approximately 40 min (4 times of 10 min), which corresponds to 280 min or 4.66 h of exposure.

**3. Results**

*3.1. Population Characterization*

Approximately 114 athletes participated in the competition, and a total of 41 athletes (33.85 ± 8.88 years) participated in this study, which represents 35.96% of the available population. Information about the origin of the disability, functional class and other characteristics is shown in Table 1.

**Table 1.** Participant characteristics.

| Sociodemographic Characteristics | *n* (%) |
|---|---|
| Manual dominance | |
| Right | 34 (83%) |
| Left | 5 (12.1%) |
| Both | 2 (4.9%) |
| Etiology of disability | |
| Congenital | 9 (22%) |
| Acquired | 32 (78%) |
| Disability diagnosis | |
| Amputation | 11 (26.9%) |
| Arthrogryposis | 3 (7.3%) |
| Difference in limb length | 1 (2.4%) |
| Spinal cord injury | 14 (34.1%) |
| Bad formation | 2 (4.9%) |
| Myelomeningocele | 4 (9.7%) |
| Polio sequelae | 5 (12.2%) |
| Hip arthrosis | 1 (2.4%) |
| Day to day assistive technologies | |
| Walking stick | 3 (7.3%) |
| Wheelchair | 23 (56.1%) |
| Crutch | 7 (17.1%) |
| Prosthesis | 4 (9.7%) |
| Bracing | 4 (9.7%) |

**Table 1.** *Cont.*

| Sociodemographic Characteristics | *n* (%) |
|---|---|
| Functional class | |
| 1.0 | 8 (19.5%) |
| 1.5 | 4 (9.7%) |
| 2.0 | 5 (12.2%) |
| 2.5 | 8 (19.5%) |
| 3.0 | 2 (4.9%) |
| 3.5 | 3 (7.3%) |
| 4.0 | 7 (17.1%) |
| 4.5 | 4 (9.7%) |
| Court position | |
| Small forward | 16 (39%) |
| Shooting guard | 13 (31.7%) |
| Point guard | 3 (7.3%) |
| Center | 9 (22%) |

*3.2. Training Profile*

Considering data from the athletes' training and preparation for the competition, we highlight that most athletes perform conditioning training (strength training, aerobic training) five times a week ($n = 13$; 31.71%), 26.83% of athletes perform conditioning training at least four times a week ($n = 11$), 19.51% of athletes perform conditioning training three times a week ($n = 8$), 7.32% of athletes perform conditioning training six days a week ($n = 3$) and two days a week ($n = 3$), 2.44% of athletes perform conditioning training only once a week ($n = 1$), and 4.88% of athletes do not perform conditioning training ($n = 2$).

Furthermore, in terms of the number of hours per training day, most athletes reported conditioning training for at least 1 h ($n = 14$; 34.15%), 21.95% for at least 2 h ($n = 9$), 12.19% for 3 h ($n = 5$) and 24.39% for 4 h ($n = 10$); only one athlete reported conditioning training for more than 4 h (2.44%), and two athletes did not perform conditioning training (4.88%).

For wheelchair basketball training (tactical and technical training), most athletes (51.22%) reported training five times a week ($n = 21$), others reported training four times a week ($n = 14$; 34.15%), and 9.76% reported training three times a week ($n = 4$). The athletes also reported training only one day a week ($n = 1$, 2.44%) and six times a week ($n = 1$; 2.44%). Considering the time per day they spent training for wheelchair basketball, the athletes reported training for at least 4 h ($n = 15$; 36.58%), 3 h ($n = 11$; 26.83%), 2 h ($n = 7$; 17.07%), 1 h ($n = 5$, 12.19%), 5 h ($n = 2$, 4.88%) and 6 h ($n = 1$, 2.44%). The data are presented in Table 2.

**Table 2.** Characterization of the training profile.

| Variable | *n* | % |
|---|---|---|
| Frequency of physical conditioning training | | |
| Not once a week | 2 | 4.88 |
| 1 time a week | 1 | 2.44 |
| 2 times a week | 3 | 7.32 |
| 3 times a week | 8 | 19.51 |
| 4 times a week | 11 | 26.83 |
| 5 times a week | 13 | 31.71 |
| 6 times a week | 3 | 7.32 |

**Table 2.** *Cont.*

| Variable | *n* | % |
| --- | --- | --- |
| Frequency of tactical and technical training | | |
| Duration of physical conditioning training | | |
| Does not condition train | 2 | 4.88 |
| 1 h a day | 14 | 34.15 |
| 2 h a day | 9 | 21.95 |
| 3 h a day | 5 | 12.19 |
| 4 h a day | 10 | 24.39 |
| More than 4 h a day | 1 | 2.44 |
| 1 time a week | 1 | 2.44 |
| 2 times a week | 0 | 0 |
| 3 times a week | 4 | 9.76 |
| 4 times a week | 14 | 34.15 |
| 5 times a week | 21 | 51.22 |
| 6 times a week | 1 | 2.44 |
| Duration of tactical and technical training | | |
| 1 h a day | 5 | 12.19 |
| 2 h a day | 7 | 17.07 |
| 3 h a day | 11 | 26.83 |
| 4 h a day | 15 | 36.58 |
| 5 h a day | 2 | 4.88 |
| 6 h a day | 1 | 2.44 |

*3.3. Characterization of Previous Injuries and Prevalence and Incidence of Injuries during Competition*

Some athletes reported not having injuries in the 12 months that preceded the competition ($n = 26$, 63.41%), and 15 athletes reported having suffered an injury in this period (36.59%). Of the athletes who were injured in this pre competition period, five needed to leave for more than 6 months (33.33%), while others reported having to leave training for 1 to 2 months ($n = 4$, 26.67%), and three reported having to leave for 3 to 4 months (20%); there was also leave for a period of 4 to 5 months ($n = 1$, 6.67%), 5 to 6 months ($n = 1$, 6.67%) and 0 to 1 month ($n = 1$, 6.67%). Of the injuries presented in the period prior to the competition, the shoulder region was the most affected ($n = 7$, 46.67%) followed by the spinal column ($n = 2$, 13.33%), fingers ($n = 2$, 13.33%), wrist ($n = 1$, 6.67%), elbow ($n = 1$, 6.67%), leg ($n = 1$, 6.67%) and knee ($n = 1$, 6.67%). When questioned about whether they sought care/treatment for these injuries, the athletes reported having sought help from physiotherapists ($n = 8$, 53.33%) and doctors ($n = 7$, 46.67%).

In consideration of epidemiological measures, the prevalence of sport injuries in the competition was 17.07% (seven injured athletes/forty-one athletes exposed), the incidence was 0.17 injuries per athlete (seven injuries/forty-one athletes), and the incidence rate was 0.03 injuries per athlete-hour (seven injuries/(forty-one athletes × 4.66 h) or four injuries per one thousand athletes-days (seven injuries/(forty-one athletes × six competition day). Of the injuries presented during the competition, the shoulder region was the most affected ($n = 4$, 57.14%) followed by the fingers ($n = 1$, 14.29%), wrist ($n = 1$, 14.29%) and mouth ($n = 1$, 14.29%). Only three of the athletes who had previously been injured were injured again during the competition in different regions of the body. The data are presented in Table 3.

**Table 3.** Characterization of pre- and during-competition injuries.

| Variable | *n* | % |
|---|---|---|
| **Precompetition injuries** | | |
| Yes | 15 | 36.59 |
| No | 26 | 63.41 |
| Leave period | | |
| Had to be away for more than 6 months | 5 | 33.33 |
| Needed to stay away from 5 to 6 months | 1 | 6.67 |
| Needed to stay away from 5 to 4 months | 1 | 6.67 |
| Needed to stay away from 4 to 3 months | 3 | 20 |
| Needed to stay away from 3 to 2 months | 0 | 0 |
| Needed to stay away from 2 to 1 months | 4 | 26.67 |
| Needed to stay away from 1 to 0 months | 1 | 6.67 |
| Region of body | | |
| Shoulder | 7 | 46.67 |
| Spine | 2 | 13.33 |
| Fingers | 2 | 13.33 |
| Wrist | 1 | 6.67 |
| Elbow | 1 | 6.67 |
| Leg | 1 | 6.67 |
| Knee | 1 | 6.67 |
| Assistance | | |
| Physiotherapists | 8 | 53.33 |
| Doctors | 7 | 46.67 |
| **Injuries during competition** | | |
| Yes | 7 | 17.07 |
| No | 34 | 82.93 |
| Region of body | | |
| Shoulder | 4 | 57.14 |
| Wrist | 1 | 14.29 |
| Fingers | 1 | 14.29 |
| Mouth | 1 | 14.29 |

*3.4. Features of Injuries during Competition*

Considering the injuries that occurred in the athletes throughout the competition, most of the injuries had a traumatic mechanism (*n* = 5, 71.43%), and by repetitive effort, there were two injuries (28.57%): one by repetitive effort of gradual onset (14.28%) and one due to repetitive effort with sudden onset (14.28%). Most injuries occurred by indirect contact with other athletes (*n* = 3, 42.86%), direct contact with other athletes (*n* = 1, 14.28%), direct contact with an object (*n* = 1, 14.28%) and no contact (*n* = 2, 28.57%). Considering the moment in which the injuries occurred, most were during the wheelchair propulsion (*n* = 4, 57.14%), passing the ball (*n* = 2, 28.57%) and throwing (*n* = 1, 14.28%). The athletes did not need to leave the competition due to injuries (*n* = 6, 85.71%) and were able to continue in the matches except for one of the athletes who had to withdraw from the match but returned in the next match (14.28%). When questioned about what could be related to the injury, the athletes reported the increase in physical demand due to the competition (*n* = 4, 57.14%) and the lack of attention from both the player himself and the opposing players (*n* = 3, 42.86%). The data are presented in Table 4. Taking into account the type of injury suffered, most were reported as contusion (*n* = 4, 57.14%) followed by dislocation (*n* = 1, 14.28%), and two injuries did not receive a medical diagnosis or the athletes were unable to answer (*n* = 2, 28.57%).

**Table 4.** Features of injuries during competition.

| Variable | *n* | % |
|---|---|---|
| **Injury mechanism** | | |
| Traumatic | 5 | 71.43 |
| Repetitive effort | 2 | 28.57 |
|    gradual onset | 1 | 14.28 * |
|    sudden onset | 1 | 14.28 * |
| **Contact** | | |
| Without contact | 2 | 28.57 |
| With contact | 5 | 71.43 |
|    Direct contact with other athletes | 1 | 14.28 * |
|    Direct contact with object | 1 | 14.28 * |
|    Indirect contact with other athletes | 3 | 42.86 * |
| **Time when the injury occurred** | | |
| Wheelchair propulsion | 4 | 57.14 |
| Passing the ball | 2 | 28.57 |
| Throwing | 1 | 14.28 |
| **Leave time** | | |
| Did not have to leave | 6 | 85.71 |
| Leave only 1 match | 1 | 14.28 |
| **Injury-related factors** | | |
| Increased physical demand | 4 | 57.14 |
| Lack of attention | 3 | 42.86 |
| **Type of Injury** | | |
| Contusion | 4 | 57.14 |
| Dislocation | 1 | 14.28 |
| Not identified | 2 | 28.57 |

* Percentage considering the total amount of 7 injuries.

## 4. Discussion

Our study aimed to identify and characterize sports injuries that occurred during a competition in the first Brazilian wheelchair basketball division. Through the analysis of our results we found that (a) the athletes who composed the sample perform a high volume of training; (b) the majority did not present injuries before the competition; (c) there was a prevalence of 17.1% of injuries during the competition, an incidence of 0.17 injuries per athlete and an incidence rate of 0.03 injuries per athlete-hour or four injuries per one thousand athletes-days; (d) and the injuries that occurred during the competition were mostly in the shoulder region, characterized as traumatic, by indirect contact with other athletes, which occurred when propelling the wheelchair, and were of low severity.

### 4.1. Training Profile and Injuries

There are important factors related to training that are closely linked to the appearance or prevention of sports injuries, such as the volume, intensity and frequency of training, as well as good technical and tactical preparation for the modality [11]. In this sense, both the technical team (coaches, physical trainers, assistants) and the medical team (doctors and physiotherapists) try to find the balance between improving an athlete's performance and preventing the athlete from acquiring an injury. There is a line of thought that defends that higher loads (internal and external) during training may increase the risk of injuries in athletes. However, evidence shows that training has a protective factor for injuries; therefore, greater loads would generate greater adaptations and lower the risk of injuries, characterizing the model "Training-Injury Prevention Paradox" [12].

In our study, most athletes did not present injuries before the competition (63.4%), demonstrating a low incidence of injuries during the training and preparation phase for the competition even if the volume and frequency of the reported training sessions were high. Most athletes reported performing conditioning training (physical and aerobic) five times a week for at least 1 h and modality training (tactical and technical) five times a week for at least 4 h. In this sense, the risk of injury without contact seems to be more related to the incorrect execution of the activity and how the training progression is carried out rather than the volume, frequency or intensity of the training. From this perspective, it is essential to determine the 'ideal point' of training stimulus, taking into account the benefits arising from training (performance improvement) through an adequate training load while balancing the negative consequences of training, such as injuries, illnesses, fatigue and overtraining [13]. Therefore, monitoring internal and external loads properly is a good ally for structuring training, optimizing gains and reducing the occurrence of injuries [12].

A study carried out with female basketball players, which aimed to determine whether functional tests can predict sports injuries in elite basketball players, had the participation of 351 players who were evaluated during the preseason period. During this assessment, evaluators used functional tests to identify players at risk of injury and found that imperfect functional movement patterns and jump landing biomechanics during preseason screening were associated with lower-extremity injuries. These results demonstrate the importance of screening for functional deficits in the training phase with a view to mitigating the onset of injuries [14]. Currently, there are no studies in the literature with wheelchair basketball, as mentioned above. The validation of functional tests for athletes who use wheelchairs and the evaluation of these athletes with these tests in preseason periods can be beneficial for the creation of specific injury prevention programs.

### 4.2. Prevalence and Incidence of Injuries during Competition

During the competition, the athletes presented low values of prevalence (17.1%), incidence (0.17 injuries per athlete) and incidence rate (0.03 injuries per athlete-hour or four injuries per one thousand athletes-days); however, it is noteworthy that, in this study, of the 114 athletes in the competition, 41 were included in the research, which represents 35.96% of the total sample. In a prospective cohort study, which was carried out during the Wheelchair Basketball World Championships 2018 (WBWC) held in Hamburg, Germany, from August 16 to 26, 2018, 100 injuries were identified, equivalent to 75.8 per 100 athletes (95% CI: 60.9–90.7) or 68.9 per 1000 athletes-days (55.4–82.4) [15]. These numbers correspond to data collected from 132 players and proportionally to the sample number of our study, which was 41 athletes; these findings demonstrate a 5.35-times higher incidence rate of injuries per 1000 athletes-days than that found in our study. This can be due to the difference in complexity between the competitions; our study was carried out in the main basketball competition in Brazil, while the competition held in Germany in 2018 was international in nature with a much larger number of athletes and a greater number of competition days.

Wheelchair basketball presents a moderate risk for sports injuries either due to the biomechanics of the modality [16,17], which needs a lot of overhead movements of the upper limbs, or by contact and collisions during matches. Mitchell et al. provide a classification of several sports modalities, relating these modalities with static and dynamic exercises and classifying them according to the intensity (low, medium and high) of the exercises (static and dynamic) [18]. In addition, they present the modalities that present a significant risk due to body collision (impact between competitors or between competitor and objects or structures) [18]. Through this classification, basketball has a high dynamic component (>70% of maximum oxygen consumption) and a static component (20–50% maximum voluntary contraction) in addition to presenting a danger of body collision [18]. In this way, basketball can be classified as a collision or contact modality.

### 4.3. Characterization of Injuries during Competition

The injuries that occurred during the competition were mostly in the shoulder region, characterized as traumatic, by indirect contact with other athletes, which occurred when propelling the wheelchair, and were of low severity. As previously mentioned, basketball is a contact modality, and for this reason, the traumatic injury mechanism is not a surprise as well as having more injuries in competition than in the training phase. Furthermore, in our study, the main region of the body affected was the shoulder region, which is in agreement with a systematic review published in 2022 that aimed to determine the epidemiological information, primary injury characteristics and body regions affected in wheelchair basketball athletes and identified that the majority of injuries in wheelchair basketball (22.1%) occur in the shoulder [19].

Shoulder injuries in wheelchair basketball may be linked to the daily demands for locomotion and transfers and, additionally, the repetitive movements of shots, passes and blocks, which mostly happen above the head, stressing the shoulder region [17]; the shoulder region is already susceptible to injuries because it is an anatomically unstable region [20]. Most of the injuries found here were not considered to have a high level of severity, so the athletes did not have to withdraw from the competition; even so, they required attention from the technical team. When not properly treated, these injuries can worsen or become recurrent injuries. In this sense, the best strategy is to design a good rehabilitation program and implement an injury prevention protocol for the entire team.

### 4.4. Preventing Injuries

Aiming for the best performance of the athlete, the screening of sports injuries is essential for the health team and, based on what is found, to outline programs for the prevention of these injuries. However, according to the results found in a study carried out by Harrington et al., which aimed to survey athletes in elite swimming, cycling and paralympic sports in the United States to better understand common injuries among athletes in each sport, determine whether injury prevention programs were being used and that involved the participation of 144 athletes (83 men and 61 women), only 24% of respondents reported having participated in an injury prevention program [21]. In the study by Harrington et al., 42% of respondents had some complaint of musculoskeletal pain; these results demonstrate the lack of well-structured injury prevention programs for athletes, considering that, due to these pain complaints, many athletes need to be absent from training and/or competitions [21].

There are some well-established injury protocols in other sports, such as FIFA 11+ [22,23], created to prevent lower limb injuries in soccer players. In addition, a shoulder injury program for goalkeepers was created, FIFA 11+ S, which presents a series of well-structured exercises for the shoulder [24]. Currently, there is no FIFA 11+ style injury prevention program in the literature for basketball athletes, much less for wheelchair basketball athletes, to be integrated as an essential part of training. However, it is already possible to find some intervention programs [17,25] that are not as structured as in the FIFA 11+ model, which can be used to mitigate the appearance of injuries. The use of the FIFA 11+ S program could be adapted for prevention programs in wheelchair sports since, for these sports in general, most injuries occur in the shoulder.

### 4.5. Limitations and Future Directions

This study has some limitations that can be addressed in future studies. In addition to training volume and frequency data, information related to the preparation of these athletes was not collected (intensity, recovery, sleep, food and presence of health professionals on the team, among others) that could be correlated with the appearance of sports injuries during competition and used to draw a profile of related risk factors. Data on the appearance of injuries were collected on the last day of competition through interviews and responses to the questionnaire used here, which may have generated a subjectivity bias in which the athletes may have overestimated or underestimated the information given.

Future directions can include a longitudinal follow-up of wheelchair basketball athletes, an analysis of the most diverse variables involved in training, the identification of injuries in the preparation and competition period to determine the main risk factors involved in the appearance of injuries and an investigation of the modulation that intervention programs in injury prevention can offer.

## 5. Conclusions

Through the results found in our study, wheelchair basketball athletes presented a low prevalence, incidence and incidence rate of injuries during the competition. The shoulder region was the most affected through a traumatic mechanism, and in terms of severity, all injuries were considered mild. Structuring training sessions with the prevention of injuries in the shoulder region in mind is essential for these athletes to perform optimally.

**Author Contributions:** Conceptualization, K.S. and J.G.; methodology, K.S. and M.M.e.S.; formal analysis, K.S.; writing—original draft preparation, K.S. and J.G.; writing—review and editing, A.C.e.S. and M.M.e.S. All authors have read and agreed to the published version of the manuscript.

**Funding:** This study was financed in part by the Coordenação de Aperfeiçoamento de Pessoal de Nível Superior—Brazil (CAPES)—Finance Code 001.

**Institutional Review Board Statement:** This research was approved by the research ethics committee of the Institute of Health Sciences of the Federal University of Pará under the number CAAE 60806322.3.0000.0018.

**Informed Consent Statement:** Informed consent was obtained from all subjects involved in the study.

**Data Availability Statement:** The data presented in this study are available on request from the corresponding author. The data are not publicly available due to preserve the privacy of the participants and teams involved.

**Acknowledgments:** We would like to thank the Brazilian Confederation of Wheelchair Basketball (CBBC) for their support throughout the research and all the teams that participated in the competition and answered the questionnaire for their support.

**Conflicts of Interest:** The authors declare no conflict of interest.

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
