# Peer review of "Identification and Characterization of Injuries during Competition in Wheelchair Basketball"

_traumacare, doi:10.3390/traumacare3020007_

Round 1
Reviewer 1 Report
The introduction does not provide relevant information on the injuries suffered by athletes with disabilities or in wheelchairs, it does not establish the relevance of the problem that they intend to study, nor does it establish research objectives.
The identification of the athletes is not correct, because they indicate a supposed level (Local/regional level, National level, International level) but they all compete in the first Brazilian wheelchair basketball division (national level).
They should review the results because some sums do not add up, for example in paragraph 99 1 105 there are only 40 participants.
Paragraph 132 to 140 should be written under the heading Injuries during competition. Nor is the specific type of injury sustained or indices of practical exposure specified.
The discussion is not well documented. The first paragraph repeats or should be included in results. (lines 159-167). Section 4.1 does not discuss the results obtained but makes general reflections on sports training and injuries. The next paragraph (179-192) repeats the results but makes a poor discussion of them.
The low participation rate of those surveyed (41 out of a total of 114) does not allow the results to be generalized, as appears to be indicated in section 4.2. Lines 209 to 2017 only apply to free basketball, excluding wheelchairs where the same instrument avoids much of the contact or risk of falls. Section 4.4 is not applicable to this study either because it does not refer to the prevention of injuries in competition, which is the object of the study.
Some of the references are not applicable or are very old and out of date.
Author Response
We appreciate the suggestions and look forward to serving you as best we can.

Reviewer 2 Report
First of all, thank you very much for the opportunity to review such an interesting manuscript.
Here are some recommendations.
Abstract
-Please review the abstract writing format.
-in the results section in the summary there are several sentences that begin with capital letters after a ";" , please correct this aspect.
Material and Methods
-Create different subsections for the "Material and Methods" section. In the same way, extend the information offered in it (how to contact the sample, statistical programs, a deeper description of the questionnaire...). Also revise the writing format.
Results
-In general, the description of the results found is good. But I would recommend that you create tables to present more clearly the information presented in subsections 3.2, 3.3 and 3.4.
Discussion and _Conclusions
-Correct and very well elaborated
-
Author Response

(The authors gave the same response as above.)

Round 2
Reviewer 1 Report
Please, you must translate lines 85-87 and 109-110 into English
Author Response
Thanks for the reviews!

Reviewer 2 Report
Abstract
-The section on conclusions in the summary has a larger font size than the other sections.
Materials and methods
-Write "2.1. characterization of the study" with a capital letter at the beginning.
-The first 2 lines of subsection 2.2. are written in Portuguese.
-The first lines of subsection 2.4. are also written in Portuguese.
Results
-In table 1 they should write "congenital" with capital letter at the beginning.
Author Response
Thanks for the reviews!
